# In Vitro Antimicrobial Activity of Contezolid Against *Mycobacterium tuberculosis* and Absence of Cross-Resistance with Linezolid

**DOI:** 10.3390/microorganisms13092216

**Published:** 2025-09-22

**Authors:** Li Wang, Jianxia Chen, Yifan He, Ruijuan Zheng, Jie Wang, Xiaochen Huang, Wei Sha, Lianhua Qin

**Affiliations:** 1Department of Tuberculosis, Shanghai Pulmonary Hospital, School of Medicine, Tongji University, Shanghai 200092, China; wangli_shph@tongji.edu.cn (L.W.);; 2Clinic and Research Center of Tuberculosis, Shanghai Pulmonary Hospital, School of Medicine, Tongji University, Shanghai 200092, China; 3Clinical and Translational Research Center, Shanghai Pulmonary Hospital, Tongji University School of Medicine, Shanghai 200433, China; 4Shanghai Key Laboratory of Tuberculosis, Shanghai Pulmonary Hospital, School of Medicine, Tongji University, Shanghai 200092, China

**Keywords:** contezolid, *Mycobacterium tuberculosis*, linezolid, drug-resistance, cross-resistance, in vitro study

## Abstract

Tuberculosis (TB) persists as a formidable global health threat, especially with the rising incidence of multidrug-resistant strains. This study aimed to evaluate the in vitro activity of contezolid, a novel oxazolidinone antibiotic, against *Mycobacterium tuberculosis* (*Mtb*) and assess potential cross-resistance with linezolid. Thirty-one *Mtb* clinical isolates (5 susceptible, 8 multidrug-resistant [MDR], 18 pre-extensively drug-resistant [pre-XDR]) were tested. Minimum inhibitory concentrations (MICs) of contezolid and linezolid were determined, along with mutation resistance frequencies. Intracellular replication inhibition in macrophages and whole-genome sequencing of resistant colonies were assessed. Cytotoxicity was evaluated via luciferase-coupled ATP assay. The MIC50 and MIC90 values of contezolid were comparable to those of linezolid. Contezolid induced higher mutation frequencies in 7 isolates. At 12 mg/L, both drugs similarly inhibited intracellular *Mtb* replication. Whole-genome sequencing revealed that the *mce3R* gene was linked to contezolid resistance, with no cross-resistance observed between two drugs. No significant cytotoxicity was observed in contezolid-treated mouse peritoneal macrophages (*p* > 0.05). Contezolid exhibits anti-*Mtb* activity, with *mce3R* potentially associated with resistance. No cross-resistance with linezolid was found.

## 1. Introduction

Tuberculosis (TB) remains a significant global health challenge, particularly with the increasing prevalence of multidrug-resistant strains [1]. Effective treatment options are crucial for controlling the spread of TB and improving patient outcomes. Linezolid is one of the important alternative drugs for rifampicin-resistant (RR-TB) or multidrug-resistant tuberculosis (MDR-TB) [2]. However, long-term use of linezolid is associated with adverse events due to peripheral neuropathy and myelosuppression and leading to poor treatment adherence and treatment discontinuation [3,4,5]. Therefore, there is an urgent need to find an antibiotic that is safer and more effective than linezolid for the management of multidrug-resistant tuberculosis (MDR-TB).

Contezolid (also known as MRX-I) is a novel oxazolidinone agent designed to possess potent antimicrobial activity and lower myelosuppression and monoamine oxidase inhibition based on the structure–activity relationship [6,7]. Preclinical and clinical investigations have demonstrated that contezolid exhibits remarkable antibacterial efficacy against a wide spectrum of Gram-positive bacteria [8,9]. Notably, it demonstrates potent activity against antibiotic-resistant strains, including methicillin-resistant *Staphylococcus aureus* (MRSA) and vancomycin-resistant *Enterococcus* (VRE) [10]. Contezolid has also shown significant antibacterial effects against Mycobacterium species [11,12]. Importantly, contezolid demonstrates superior safety profiles compared to linezolid, evidenced by reduced myelosuppressive potential and lower monoamine oxidase inhibitory activity, as supported by preclinical and clinical data [13,14].

The preliminary evidence currently available has revealed the potential utility of contezolid in the treatment of TB. Deepak Almeida et al. [15] used the BALB/c murine model of tuberculosis and found that the MIC of contezolid was similar to that of linezolid. Li et al. [16] reported a case in which a patient with tuberculosis infection after allogeneic hematopoietic stem cell transplantation was successfully treated with contezolid. However, more laboratory and clinical data are needed to confirm the value of contezolid in the management of TB. The present study aimed to determine the in vitro activity of contezolid against clinical isolates of *Mtb*, which were identified as susceptible, multidrug-resistant (MDR), or pre-extensively drug-resistant (pre-XDR). This study also investigated the intracellular anti-*Mtb* activity, cross-resistance with linezolid, and cytotoxicity of contezolid compared with linezolid.

## 2. Materials and Methods

### 2.1. Bacterial Strains and Cultures

China General Microbiological Culture Collection Center provided the reference strain H37Rv (ATCC 27294). The clinical strains were isolated from sputum specimens of patients enrolled in the National Twelve Five-year Science and Technology Major Project of China (from the year of 2014 to 2018, and 7 provinces), and simultaneously identified by growth characteristics, colony morphology, and growth in the presence of PNB (*p*-nitrobenzoic acid) and TCH (thiophene-2-carboxylic acid hydrazide), and 16SrRNA sequencing. Thirty-one isolates of *Mtb* were selected randomly from the laboratory stock of clinical isolates. All test strains were grown in Middlebrook 7H9 broth (Difco/Becton Dickson, Franklin Lakes, NJ, USA) supplemented with 10% ADC [5% bovine serum albumin (BSA, ≥98% purity, Sigma-Aldrich, St. Louis, MO, USA), 2% dextrose (≥99% purity, Sigma-Aldrich), 5% catalase (≥99% purity, Sigma-Aldrich)] and 0.05% Tween-80 (≥99% purity, Sigma-Aldrich) or Middlebrook 7H10 agar (Difco) supplemented with 10% ADC and antibiotic supplements as required. The antibiotics used were 50 μg/mL or 75 μg/mL hygromycin B (≥95% purity, Thermo Fisher Scientific, Waltham, MA, USA) and 50 μg/mL kanamycin (≥99% purity, Sigma-Aldrich) [17].

The study was approved by the Ethics Committee of Tongji University affiliated Shanghai Pulmonary Hospital (No. K20-162Z). The study was conducted in accordance with the principles of the Declaration of Helsinki and the ethical and biosafety agreements of the institution.

### 2.2. Susceptibility Testing

BACTEC MGIT 960 system was used to test the susceptibility of *Mtb* isolates to anti-TB drugs, including streptomycin, isoniazid, rifampin, ethambutol, levofloxacin, moxifloxacin, and amikacin [18,19]. The minimum inhibitory concentrations (MICs) of tested drugs were determined by a modified microdilution method in 96-well plates containing Middlebrook 7H9 liquid medium supplemented with 0.25% glycerol, 10% oleic acid-albumin-dextrose-catalase, and 0.05% Tween-80, as described by Qin L and Kumar et al. [17,20]. The final concentrations of streptomycin, rifampin, ethambutol, and amikacin were designed as 32, 16, 8, 4, 2, 1, 0.5, 0.25 mg/L, isoniazid and moxifloxacin as 8, 4, 2, 1, 0.5, 0.25, 0.125, 0.06 mg/L, and levofloxacin as 16, 8, 4, 2, 1, 0.5, 0.25, 0.125 mg/L. Contezolid (MicuRx Pharmaceuticals, Shanghai, China, ≥99% HPLC purity) and linezolid (Airsea Pharmaceuticals, Taizhou, China, ≥99% HPLC purity) were prepared via two-fold serial dilution (final concentrations: 16–0.03 mg/L). Replicates and reference strains were included in each run.

### 2.3. Induced Mutation Frequencies

The reference strain H37RV and clinical isolates (Y2, Y16, Y23, Y26, Y48, and Y117) were cultured to log phase and inoculated onto Middlebrook 7H10 plates containing contezolid or linezolid (2-fold MIC). The inoculum size of the test strains was 10^9^, 10^10^, or 10^11^ CFU. The plates were incubated at 37 °C to observe the growth of mycobacterial colonies weekly. A negative result was reported if no growth was found after culture for 8 weeks. The frequency of mutational resistance was calculated as the ratio between the post-exposure CFU values and the pre-exposure CFU values. If no growth was observed on the drug-containing plate, then the frequency of mutational resistance was “less than 1” divided by the CFU inoculated on the drug-containing plate. Positive clones were harvested from the Middlebrook 7H10 plates for analysis of resistant mutations.

### 2.4. Inhibition of Intracellular Mtb Replication

Primary mouse peritoneal macrophages were prepared and cultured as previously described [21]. H37Rv and Y-144 were grown to the mid-log phase in the Middlebrook 7H9 broth medium. More preparation details can be seen in Kong and Ge (2008) [21]. Adherent monolayer peritoneal macrophages were infected by adding the bacterial preparations at an MOI of 2. At 3 h postinfection, the cells were washed with PBS to remove bacteria. The two drugs contezolid and linezolid were added into the cell culture to the final concentration of 6 mg/L or 12 mg/L. After 12 h, the drugs were washed out thoroughly to remove from the culture. Cells were lysed, and the cell lysates were flattened on 7H10 solid plates and incubated at 37 °C for 3–4 weeks. CFU were counted and the results were expressed as the inhibitory rate.

### 2.5. Selection of Highly Oxazolidinone-Resistant Strains

All the positive clones harvested from mutation tests were inoculated into Middlebrook 7H9 liquid medium containing contezolid or linezolid at a concentration of 4-fold MIC. The mixture was incubated at 37 °C for 2 weeks to obtain high-resistant strains for determination of mutation. The positive clones at the mid-log phase were diluted to the density of OD600 0.25 for MIC determination. Contezolid and linezolid were prepared to final concentrations of 32, 16, 8, 4, 2 mg/L. The highly resistant strains with a ≥8-fold MIC increase were selected for whole-genome sequencing analysis.

### 2.6. Cross-Resistance Analysis of Oxazolidinone-Resistant Strains

The susceptibility of contezolid-resistant strains to linezolid and the susceptibility of linezolid-resistant strains to contezolid were tested in a 96-well microplate containing two-fold diluted concentrations of contezolid or linezolid (16, 8, 4, 2, 1, 0.5, 0.25, 0.125, 0.06, 0.03 mg/L). The results were analyzed to examine the potential cross-resistance of oxazolidinone-resistant strains.

### 2.7. Whole-Genome Sequencing Analysis

The genomes of highly oxazolidinone-resistant strains were extracted using a genetic sample kit (HiPure Bacterial DNA Kit, Magen Biotech Co., Ltd. Guangzhou, China). A 150 bp paired-end shotgun whole-genome sequencing was performed on the Illumina (San Diego, CA, USA) NovaSeq platform using the Nextera XT DNA sample preparation kit (Illumina, San Diego, CA, USA) according to the manufacturer’s instructions [22]. Prior to analysis, raw reads were processed with Trimmomatic v0.39 to remove adapter sequences and low-quality bases, employing a sliding window trimming strategy (window size: 4, quality threshold: 20) and retaining reads with a minimum length of 50 bp. All the genome data has been uploaded to NODE (National Omics Data Encyclopedia, https://www.biosino.org/node/ accessed on 11 August 2025), and the project ID is OEP004777.

### 2.8. Mouse Peritoneal Macrophage Isolation and Cytotoxicity Detection

Mouse peritoneal macrophages were prepared as described below. Six-week-old male C57BL/6 mice (Cyagen Biosciences, Guangzhou, China) were anesthetized with isoflurane and subsequently sacrificed by cervical dislocation for peritoneal macrophage isolation. The abdominal skin was disinfected with 75% ethanol, and a small incision was made to expose the peritoneum. A total of 5 mL pre-warmed sterile phosphate-buffered saline (PBS, pH 7.4; Gibco/Thermo Fisher Scientific, Waltham, MA, USA) was slowly injected into the peritoneal cavity using a 21-gauge needle. The abdomen was gently massaged for 1–2 min to promote detachment of macrophages from the peritoneal lining. The PBS containing suspended macrophages was then aspirated with a syringe and transferred into a 15 mL centrifuge tube. The cell suspension was centrifuged at 400× *g* for 5 min at 4 °C. After discarding the supernatant, the cell pellet was resuspended in RPMI 1640 medium (Gibco) supplemented with 10% fetal bovine serum (FBS; Gibco/Thermo Fisher Scientific). Cell counting was performed using a hemocytometer, and the cell density was adjusted to 1 × 10^6^ cells/mL. The cell suspension was seeded into 96-well plates at a volume of 100 μL per well and incubated at 37 °C in a humidified atmosphere with 5% CO_2_. Following 2–3 h of incubation, non-adherent cells were gently removed by washing the wells three times with pre-warmed PBS, retaining only the adherent peritoneal macrophages for subsequent experiments.

The cytotoxicity of contezolid and linezolid was evaluated using the luciferase-coupled ATP assay [23]. Mouse peritoneal macrophages were exposed to varying concentrations of contezolid and linezolid (50, 25, 12.5, 6.3, 3.2, 1.6, 0.8, and 0.4 mg/L) for 72 h. Cytotoxicity was assessed using the Enhanced ATP Assay Kit (Beyotime Biotechnology, Shanghai, China, Cat. No. S0027) according to the manufacturer’s instructions. For sample lysis, 120 μL of cell lysis buffer was added to each well, and the process was performed at 4 °C. The ATP detection reagent was diluted with ATP detection reagent diluent at a ratio of 1:4. After adding the ATP detection working solution, the mixture was incubated at room temperature for 3–5 min. Subsequently, the cell supernatant was added to each well, mixed thoroughly, and the plate was immediately loaded onto a luminometer for detection. Culture wells containing cells without drug treatment served as controls. Relative luminescence unit (RLU) values were measured using a luminometer. Relative cell viability was calculated as the ratio of RLU values in experimental wells to those in control wells, which was used to reflect the degree of cytotoxicity.

### 2.9. Statistical Analysis

The MIC data of contezolid and linezolid against *Mtb* were presented as MIC range, MIC_50_, and MIC_90_. The mutation frequencies and cross-resistance analysis were analyzed with descriptive statistics. The inhibition rates of intracellular *Mtb* replication and the results of the lactate dehydrogenase leakage assay were compared between contezolid and linezolid using a *t*-test. If the assumptions of normality were not met, non-parametric statistical tests were used as alternatives. The difference was statistically significant when *p* < 0.05. Statistical analysis was conducted with SPSS 19.0 (IBM Corp, Armonk, NY, USA).

## 3. Results

### 3.1. Resistance Profiles of Mtb Isolates

In total, 31 *Mtb* isolates were randomly selected from the clinical isolate repository, including 5 susceptible, 8 multidrug-resistant (MDR), and 18 pre-extensively drug-resistant (pre-XDR) strains (Table 1). All the MDR and pre-XDR isolates showed high-level isoniazid resistance (MIC > 1.0 mg/L). Most (77.4%, 24/31) of the isolates were highly resistant to rifampicin (MIC ≥ 4.0 mg/L), including 20 isolates with very high-level rifampicin resistance (MIC > 32 mg/L). All 18 pre-XDR isolates were resistant to levofloxacin, including 4 with high-level resistance (MIC ≥ 8.0 mg/L), and 16 were moxifloxacin resistant, among which 4 isolates with MIC ≥ 2.0 mg/L showed high-level moxifloxacin resistance (Table 1).

### 3.2. MICs of Contezolid and Mutation Frequencies of Mtb

The MICs of contezolid and linezolid against thirty-one isolates are shown in Table 1. For susceptible, MDR, and pre-XDR isolates, the MIC_50_ values of contezolid were 1 mg/L, 1 mg/L, and 1 mg/L, respectively, with corresponding MIC_90_ values of 2 mg/L, 2 mg/L, and 16 mg/L (Table 2). These values were comparable to those of linezolid (MIC_50_: 0.5, 1, 1 mg/L; MIC_90_: 1, 1, 16 mg/L). Pre-XDR isolates exhibited the highest resistance to both drugs (MIC_90_ = 16 mg/L for both).

Wild-type (H37Rv) and clinical isolates of susceptible-*Mtb*, MDR-*Mtb*, and pre-XDR-*Mtb* (2 each) were randomly selected to determine mutation frequencies induced by contezolid and linezolid. The mutation frequencies (2×) induced by linezolid were 1.1 × 10^−10^ to 6.3 × 10^−9^, significantly lower than those induced by contezolid (2.44 × 10^−8^ to 7.6 × 10^−8^) (Table 3).

### 3.3. Inhibition of Intracellular Mtb Replication

H37Rv and one MDR isolate (Y-144) with high-level resistance to isoniazid and rifampicin were used to assess contezolid’s effect on intracellular *Mtb* replication. Contezolid exhibited dose-dependent inhibition of the replication of intracellular *Mtb*. For both H37Rv and Y-144, 12 mg/L contezolid yielded higher inhibition rates than 6 mg/L. At a concentration of 6 mg/L, contezolid and linezolid resulted in inhibitory rates of 50.3% ± 19.5% vs. 52.6% ± 9.9% (*p* = 0.03) for H37Rv, and inhibitory rates of 48.9% ± 27.7% vs. 44.2% ± 6.2% (*p* = 0.05) for Y-144. At the concentration of 12 mg/L, the two drugs showed comparable inhibitory rates on the replication of intracellular *Mtb* (78.1% ± 9.0% and 71.7% ± 2.8% (*p* = 0.14) for H37Rv, 61.3% ± 6.2% and 56.7% ± 1.4% (*p* = 0.11) for Y-144, Figure 1).

### 3.4. Mutations Associated with Contezolid Resistance and Cross-Resistance with Linezolid

Two susceptible isolates (Y23 and Y26) and one pre-XDR isolate (Y117) were incubated with and induced by contezolid or linezolid. The highly oxazolidinone-resistant clones were selected to investigate resistance-associated mutations. The contezolid-resistant clones derived from the original strain Y23 were named C39-Y23 and C41-Y23. The linezolid-resistant clones derived from the original strain Y23 were named L43-Y23. The fold change values of MIC were calculated for the strains to confirm their resistance phenotype. The MIC of contezolid against the 3 clinical isolates increased 16- to 64-fold after induction, while the MIC of linezolid increased 8- to 32-fold after induction (Table 4). The drug resistance phenotype was tested to explore whether the contezolid-resistant strains were cross-resistant to linezolid. The MIC of linezolid was 0.5 mg/L against 4 contezolid-resistant strains and 1 mg/L against 2 contezolid-resistant strains, indicating that they were not cross-resistant to linezolid. However, the elevated MICs of contezolid against linezolid-resistant mutants suggested that the linezolid-resistant strains could be contezolid-resistant (Table 4).

Clones derived from susceptible isolates Y23 and Y26 harbored more mutations than those from pre-XDR isolate Y117 (Table 4, Appendix A). Seventeen mutated genes were identified in susceptible-derived clones, including 10 previously reported (e.g., *Rv0197*, *rplC*, *ceoB*) and 7 novel genes. Conversely, most mutations in pre-XDR-derived clones were previously documented.

Among the reported genes, *ceoB*, integrated mobile genetic elements, and *Rv2082* were mutated in both contezolid- and linezolid-resistant strains. Three mutated genes *mce3R*, *mas*, and *esxM* were found in two or three contezolid-resistant strains. No direct evidence was available for the involvement of *mas* and *esxM* in oxazolidinone-resistance. These two genes were related to the virulence of *Mtb* and the extent of extrapulmonary dissemination. The only gene with mutations identified in the contezolid-resistant strains derived from all of the 3 original strains was *mce3R*. Four mutants harbored point mutations of *mce3R* which resulted in a change in the amino acid sequence. Notably, no point mutation was detected in C41-Y23 and C14-Y26, and heterogeneous insertion or deletion mutations were identified in the sequence analyses. The percentage of heterogeneous mutations was 90.77% in C41-Y23 and 22.60% in C14-Y26 (Table 4). Both mutations resulted in a frameshift, which probably caused the change in gene function.

### 3.5. Cytotoxicity Induced by Contezolid

The cytotoxicity of contezolid was evaluated in mouse peritoneal macrophages using the luciferase-coupled ATP assay (Figure 2). As shown in Figure 2, the relative viability of mouse peritoneal macrophages treated with contezolid or linezolid exhibited no significant decrease across all tested concentrations (0.4 to 50 mg/L, *p* > 0.05). These findings demonstrated that neither contezolid nor linezolid induced significant cytotoxicity in mouse peritoneal macrophages within the concentration range of 0.4 to 50 mg/L, thereby verifying their safety. Furthermore, at each tested concentration, the cell viability following contezolid treatment was consistently higher than that after linezolid treatment. Despite the absence of statistically significant differences (*p* > 0.05), this consistent trend implies that contezolid may have a superior safety profile compared to linezolid.

## 4. Discussion

Contezolid has potent antimicrobial activity with an improved safety profile compared with linezolid, especially in adverse events related to myelosuppression and monoamine oxidase inhibition [7]. It is reported that oral contezolid was well tolerated at 800 mg every 12 h for up to 28 days, and discontinuation of treatment due to adverse reactions was not identified [14].

The activity of contezolid against Gram-positive bacteria has been tested in a large pool of clinical isolates [24]. Contezolid also showed activity against *Mtb* in vitro, with an MIC_50/90_ of 0.5/1 mg/L against 20 susceptible or isoniazid-resistant isolates [11]. In the present study, the MIC_50/90_ values of contezolid against MDR and pre-XDR isolates were higher than those in the previous report [11], which was probably due to the sample size and more MDR and pre-XDR clinical isolates were selected. The activity of contezolid in inhibiting intracellular *Mtb* replication is also critical because *Mtb* is an intracellular pathogen infecting alveolar macrophages. Contezolid demonstrated similar inhibitory rates on intracellular *Mtb* replication compared to linezolid at 12 mg/L (Figure 1).

In our study, several contezolid-induced strains produced higher mutation frequencies than linezolid-induced strains (Table 3). Gumbo et al. [25] reported that the average frequencies of resistance mutations induced by first-line anti-TB drugs isoniazid, rifampicin, and ethambutol were 2.6 × 10^−8^, 2.3 × 10^−10^ and 1 × 10^−7^, respectively. The frequency of mutation induced by contezolid (2.44 × 10^−8^ to 7.6 × 10^−8^) was comparable to those induced by the first-line anti-TB drug, isoniazid (2.6 × 10^−8^). Anti-TB monotherapy will quickly lead to the emergence of drug resistance and treatment failure. Multidrug combination therapy is essential to reduce the risk of developing drug-resistant mutants and improve treatment success [26].

Three mutated genes were identified in contezolid-induced *Mtb* strains. The gene encoding mycocerosic acid synthase (*mas*) is involved in the synthesis of dimycocerosyl phthiocerol [27], which is one of the characteristic virulence factors of *Mycobacterium* [28]. Another gene *esxM* is involved in macrophage functional disorders and aggravation of extrapulmonary dissemination [29]. The mutations of *esxM* in the two mutants were nonsense. Although these two mutated genes (*mas* and *esxM*) were not associated with drug resistance directly, their mutations under drug pressure probably implied the evolution of strains or acquired compensatory mutations. A previous study reported that mutations in *mce3R* were common in contezolid-resistant isolates (93.4%, 99/106) [30]. In our study, the point mutations in *mce3R* were detected in 4 contezolid-resistant strains. These mutations were all nonsynonymous but not reported in the study of Pi et al. [30]. Our study provides new data on the existing mutation spectrum of the *mce3R* gene, including two heterogeneous frameshift mutations. It is now unclear how *mce3R* leads to contezolid resistance. It is speculated that *mce3R* may promote contezolid degradation by regulating monooxygenase [30]. The three mutated genes were not identified in linezolid-resistant strains, which may explain the absence of cross-resistance with linezolid. This hypothesis was partly confirmed by the preliminary susceptibility testing data (Table 4). The MIC of linezolid against contezolid-resistant strains was 0.5 to 1 mg/L, and no cross-resistance with linezolid was observed. However, the linezolid-resistant strains showed high MIC values of contezolid (Table 4), indicating that they were contezolid-resistant. It is worth considering this phenomenon in clinical practice. The functions of the other 7 mutated genes have not been reported previously. These genes may be involved in the evolution of oxazolidinone resistance through acquiring compensatory mutations.

The chemical structure modification has improved the safety profile of contezolid. In the present study, the cytotoxicity assay indicated minimal cytotoxicity of contezolid at a high dose of 50 mg/L in vitro. Two clinical trials in healthy adults confirmed that oral contezolid 800 mg q12h was well-tolerated [14,31]. Preliminary clinical experience has provided evidence of its safety that one patient with tuberculous pleurisy received contezolid plus cycloserine due to thrombocytopenia induced by linezolid for 29 days and another patient with tuberculous meningoencephalitis received regimens including contezolid for 11 weeks [32,33].

The fight against TB requires a multifaceted approach that integrates scientific advancements, policy initiatives, and community engagement. By addressing the social determinants of TB, improving access to effective treatments, and investing in innovative research, we can move closer to the goal of global TB elimination. Future research should also explore the potential of personalized medicine and novel technologies, such as bioinformatics and AI, to tailor interventions to individual patient profiles.

To the best of our knowledge, this is the first time to study the frequency of resistance mutation in clinical isolates of *Mtb* induced by contezolid. The results provide more points that *mce3R* mutation may be related to inducible contezolid resistance.

## 5. Limitations

Firstly, the number of MDR and pre-XDR isolates is relatively small, making it insufficient to accurately represent the prevalence of *Mtb* in clinical practice. Additionally, the availability of experimental materials constrained our ability to conduct a broader range of tests on the isolated strains. Future studies with larger sample sizes and diverse populations are needed to validate our results and explore additional aspects of TB pathogenesis and management. Furthermore, the association between the *mce3R* gene and contezolid resistance requires further characterization to better understand its role in resistance mechanisms.

## 6. Conclusions

In the present study, we investigated the in vitro activity of contezolid against MDR-*Mtb* and pre-XDR-*Mtb*. Our results demonstrate that contezolid exhibits potent activity against both MDR-*Mtb* and pre-XDR-*Mtb*, including the inhibition of intracellular *Mtb* replication. This suggests that contezolid is a valuable addition to the current arsenal of anti-TB drugs, particularly for treating drug-resistant strains. Additionally, the gene *mce3R* was probably associated with contezolid-resistance.

## Figures and Tables

**Figure 1 microorganisms-13-02216-f001:**
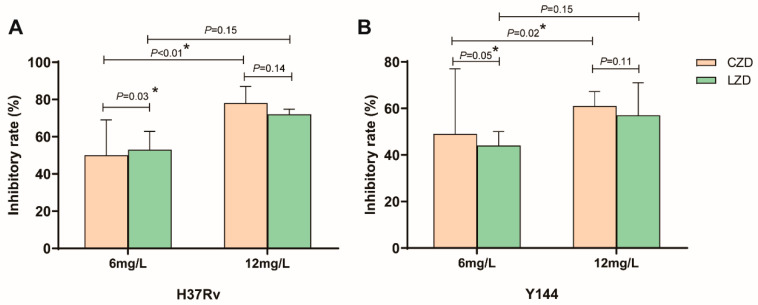
The inhibitory rate of contezolid and linezolid on intracellular *Mtb* replication. (**A**) Comparison of the inhibitory effects of the two drugs on H37Rv replication at concentrations of 6 mg/L and 12 mg/L. (**B**) Comparison of the inhibitory effects of the two drugs on Y-144 (one MDR-TB clinical isolate) replication at concentrations of 6 mg/L and 12 mg/L. Inhibitory rate = (CFU in the control group-CFU in the drug treatment group)/CFU in control group × 100%. * reprensents the *p*-value < 0.05.

**Figure 2 microorganisms-13-02216-f002:**
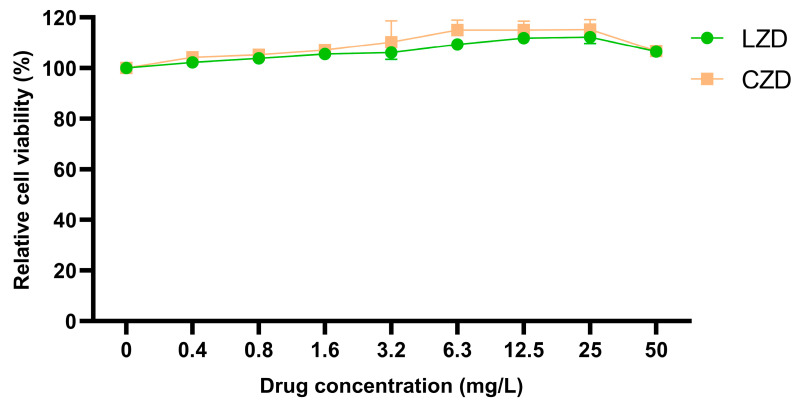
Luciferase-coupled ATP assay to examine the cytotoxicity of contezolid and linezolid. Mouse peritoneal macrophages were exposed to contezolid or linezolid at different concentrations for 72 h. Each data column and bar represent the mean value of relative cell viability (%) ± SD.

**Table 1 microorganisms-13-02216-t001:** MICs of anti-tuberculosis drugs against 31 *Mtb* isolates.

*Mtb* (n = 31)	Minimum Inhibitory Concentration (mg/L)								
	INH	RIF	ETB	STM	LFX	MOX	AMK	LZD	CZD
Susceptible (n = 5)									
Y23	≤0.06	≤0.25	0.5	≤0.25	≤0.125	≤0.06	0.5	0.5	0.5
Y26	≤0.06	0.25	0.25	≤0.25	≤0.125	≤0.06	0.5	0.5	0.5
Y82	≤0.06	≤0.25	0.5	≤0.25	≤0.125	≤0.06	0.5	1	2
Y100	≤0.06	≤0.25	≤0.25	≤0.25	≤0.125	≤0.06	0.5	1	2
Y161	≤0.06	≤0.25	0.5	≤0.25	≤0.125	≤0.06	0.5	0.5	1
MDR (n = 8)									
Y2	4 *	4 *	2 *	8 *	≤0.125	≤0.06	0.5	1	1
Y17	2 *	>32 *	2 *	0.5 *	0.25	0.25 *	2 *	0.5	1
Y48	2 *	>32 *	4 *	>32 *	≤0.125	≤0.06	0.5	1	1
Y111	4 *	4 *	2 *	>32 *	≤0.125	≤0.06	>32 *	2 *	2
Y114	4 *	>32 *	1 *	8 *	≤0.125	≤0.06	0.5	1	1
Y144	8 *	>32 *	1 *	≤0.25	≤0.125	≤0.06	1 *	1	4 *
Y182	>8 *	4 *	2 *	>32 *	≤0.125	0.125	0.5	1	1
Y218	4 *	>32 *	1 *	>32 *	≤0.125	≤0.06	1 *	1	2
pre-XDR (n = 18)									
Y11	2 *	>32 *	1 *	>32 *	2 *	0.25 *	4 *	0.5	0.5
Y61	>8 *	1 *	8 *	>32 *	8 *	2 *	0.5	4 *	4 *
Y16	2 *	>32 *	4 *	>32 *	4 *	1 *	>32 *	0.5	1
Y62	8 *	2 *	8 *	>32 *	4 *	0.25 *	0.5	16 *	16 *
Y88	4 *	>32 *	4 *	>32 *	4 *	1 *	0.5	8 *	0.5
Y89	>8 *	>32 *	2 *	>32 *	8 *	0.5 *	4 *	2 *	2
Y105	4 *	>32 *	4 *	>32 *	2 *	0.5 *	4 *	1	1
Y109	>8 *	>32 *	2 *	>32 *	2 *	0.5 *	4 *	1	1
Y117	8 *	>32 *	1 *	0.5 *	2 *	0.5 *	1 *	0.5	1
Y125	>8 *	>32 *	4 *	32 *	4 *	2 *	4 *	16 *	>16 *

INH: Isoniazid (first-line anti-TB drug); RIF: Rifampicin (first-line anti-TB drug); ETB: Ethambutol (first-line anti-TB drug); STM: Streptomycin (second-line anti-TB drug); LFX: Levofloxacin (fluoroquinolone antibiotic); MOX: Moxifloxacin (fluoroquinolone antibiotic); AMK: Amikacin (aminoglycoside antibiotic); LZD: Linezolid (oxazolidinone antibiotic); CZD: Contezolid (novel oxazolidinone antibiotic); MDR: Multidrug-resistant (resistant to at least INH and RIF); pre-XDR: Pre-extensively drug-resistant (resistant to INH, RIF, and at least one fluoroquinolone). * Represents resistance to the antibiotic.

**Table 2 microorganisms-13-02216-t002:** MIC_50_ and MIC_90_ values of contezolid and linezolid against susceptible, MDR, and pre-XDR Isolates.

Drug	Susceptible		MDR		Pre-XDR	
	MIC_50_	MIC_90_	MIC_50_	MIC_90_	MIC_50_	MIC_90_
Contezolid	1	2	1	2	1	16
Linezolid	0.5	1	1	1	1	16

MIC_50_ and MIC_90_ values are presented in mg/L.

**Table 3 microorganisms-13-02216-t003:** Frequency of mutational resistance induced by contezolid and linezolid.

Isolates	Phenotype	MIC (mg/L)		Mutation Frequency Induced by (2×)	
		CZD	LZD	CZD	LZD
H37Rv	Wild type	1	0.5	4.9 × 10^−8^	1 × 10^−9^
Y23	Susceptible	0.5	0.5	5.64 × 10^−8^	5 × 10^−10^
Y26	Susceptible	0.5	0.5	4.32 × 10^−8^	5 × 10^−10^
Y2	MDR	1	1	2.44 × 10^−8^	2 × 10^−10^
Y48	MDR	1	1	5.3 × 10^−8^	1.1 × 10^−10^
Y117	pre-MDR	1	0.5	7.6 × 10^−8^	1.1 × 10^−10^
Y16	pre-XDR	1	0.5	2.94 × 10^−8^	6.3 × 10^−9^

CZD, contezolid; LZD, linezolid; MDR, multidrug-resistant; pre-XDR, pre-extensively drug-resistant.

**Table 4 microorganisms-13-02216-t004:** The changes in CZD and LZD MICs in mutants and *mce3R* mutations in CZD-resistant mutants.

Mutants	MIC of CZD (mg/L)	Fold Change in CZD MIC	MIC of LZD (mg/L)	Fold Change in LZD MIC	Gene Mutations Referenced Original Strain (Reported)	Mutations in *mce3R*	
						Mutation	Deduced Amino Acid
L43-Y23	16	32	8	16	*Rv0197* ^a^, *rplC*, *ceoB*, integrated mobile genetic element		
C39-Y23	32	64	0.5	1	*dsbF*, integrated mobile genetic element	G838A	E280K
C41-Y23	32	64	0.5	1	*aroG*, *ceoB*, integrated mobile genetic element	c.2206797_2206798 ins GCCATCG	frameshift
L4-Y26	16	32	16	32	*rplC*		
C12-Y26	16	32	1	2	*mas*	G506A	G169D
C14-Y26	8	16	1	2	*pks5*, *mas*, *Rv2971*	c.2206456_2206462delGCCTCGC	frameshift
L28-Y117	32	32	4	8	*rrl, Rv2082*		
C31-Y117	32	32	0.5	1	*esxM* ^b^, *Rv2082*	G734A	G245D
C33-Y117	32	32	0.5	1	*esxM* ^b^, *Rv2082*	G815A	R272H

a. mutated into stop codon. b. nonsense mutation.

## Data Availability

The original contributions presented in this study are included in the article/Appendix A. Further inquiries can be directed to the corresponding authors.

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
