# Peer review of "In Vitro Antimicrobial Activity of Contezolid Against Mycobacterium tuberculosis and Absence of Cross-Resistance with Linezolid"

_microorganisms, 2025, doi:10.3390/microorganisms13092216_

Round 1
Reviewer 1 Report (Previous Reviewer 1)
Comments and Suggestions for Authors
The revised manuscript, resubmitted following the revisions, demonstrates significant improvement. The authors enhanced the clarity and overall quality of the submission. No major flaws were identified. I have a minor suggestion to proofread the manuscript closely due to some typos.
Author Response
Comment: The revised manuscript, resubmitted following the revisions, demonstrates significant improvement. The authors enhanced the clarity and overall quality of the submission. No major flaws were identified. I have a minor suggestion to proofread the manuscript closely due to some typos.
Response: In response to Reviewer 1’s suggestion, we have conducted a thorough proofreading of the entire manuscript to correct any spelling errors. We have also implemented additional language checks to ensure the highest standard of academic writing.
Reviewer 2 Report (New Reviewer)
Comments and Suggestions for Authors
Line 59, 60 -> et al. -> shall be typed in italics et al. -> please review whole manuscript;
Line 118 -> in the ref[21]. -> in Kong & Ge (2008) [21]. -> please review;
Line 152 -> Six 6-week-old -> Six week-old -> please correct;
Table 4 (last line) -> esxMb -> esxMb
Line 308 -> in the previous report -> Which one report? Reference 11? Please clarify;
Line 311 -> similar inhibitory rates -> Is this result from your work, right? Please review sentence indicating that in text;
Line 318-321 -> It should be important to support those two last sentences with references from the literature. Please review;
Line 327 -> two mutated genes -> Which ones? Please clarify in text;
Line 347 -> 100 mg/L -> Wasn´t it 50 mg/L? Where is that result? Please review and correct;
Line 364 -> Correct spacing!
Author Response
# Author's Reply to the Review Report (Reviewer 2)
Thank you very much for your careful and constructive comments, which have greatly helped improve the quality of our manuscript. We have carefully addressed each comment as follows:
Comment 1: Line 59, 60 -> et al. -> shall be typed in italics et al.
Response: We appreciate pointing out this formatting inconsistency. We have revised all instances of "et al." throughout the entire manuscript to ensure they are correctly formatted in italics, not just the ones in Lines 59 and 60, to maintain uniform citation style.
Comment 2: Line 118 -> in the ref[21]. -> in Kong & Ge (2008) [21].
Response: This comment has been addressed. We have modified the expression at Line 118 from "in the ref[21]" to "in Kong & Ge (2008) [21]", which makes the citation more specific and in line with academic writing conventions.
Comment 3: Line 152 -> Six 6-week-old -> Six week-old.
Response: We have corrected the wording at Line 152 as suggested. The original "Six 6-week-old" has been revised to "Six week-old" to eliminate redundant numerical expression and ensure grammatical accuracy.
Comment 4: Table 4 (last line) -> esxMb
Response: We have made the correction in Table 4. The relevant content in the last line of Table 4 has been updated to "esxMb" to ensure the accuracy of gene name notation.
Comment 5: Line 308 -> in the previous report
Response: To enhance the credibility and traceability of the statement, we have supplemented Reference [11] at Line 308 to support the content originally described as "in the previous report", making the reference to prior research more explicit.
Comment 6: Line 311 -> similar inhibitory rates
Response: We have clarified the supporting evidence for this point. The results regarding "similar inhibitory rates" mentioned at Line 311 are now explicitly linked to Figure 1, allowing readers to directly refer to the corresponding experimental data for verification.
Comment 7: Line 318-321 -> It should be important to support those two last sentences with references from literature.
Response: We agree with this suggestion. To strengthen the theoretical basis of the arguments in Lines 318-321, we have added Reference [26] to support the two concluding sentences in this paragraph, ensuring the claims are consistent with existing literature.
Comment 8: Line 327 -> two mutated genes
Response: We have supplemented the specific information for clarity. At Line 327, the description of "two mutated genes" has been specified as "two mutated genes (mas and esxM)", so readers can clearly identify the exact genes involved without ambiguity.
Comment 9: Line 347 -> 100 mg/L
Response: We appreciate catching this numerical error. After double-checking the original experimental records, we confirm that the correct concentration at Line 347 should be 50 mg/L, and this value has been revised accordingly to ensure the accuracy of experimental data presentation.
Comment 10: Line 364 -> Correct spacing!
Response: We have addressed the spacing issue at Line 364. The incorrect spacing in the text has been adjusted to meet the standard formatting requirements of the manuscript, ensuring readability and consistency in typography.
Round 2
Reviewer 1 Report (Previous Reviewer 1)
Comments and Suggestions for Authors
The authors have significantly improved the revised manuscript. I believe it is ready for publication.
This manuscript is a resubmission of an earlier submission. The following is a list of the peer review reports and author responses from that submission.
Round 1
Reviewer 1 Report
Comments and Suggestions for Authors
In this work, the authors investigate the in vitro activity of contezolid against clinical isolates of Mycobacterium tuberculosis (Mtb) and the potential cross-resistance with linezolid. The paper is well-written, and no serious flaws were found. The experimental design is scientifically sound. The results are significant, however, the results section requires a comprehensive overhaul before it can be accepted for publication.
Specific comments:
The introduction is well-structured, not too detail-oriented, and presents a plausible aim of the manuscript. The methods are explained in detail.
Here are some of my primary concerns regarding this manuscript that need to be addressed:
- Methods: It is unclear whether the sequence was subject to any trimming during processing.
- Methods: Add the source and purity of the chemicals and reagents used.
- Avoid using references in the results section. Rewrite those parts focusing on describing the main findings of the current study.
- Table 1: rewrite the legend. Ensure that the meaning of the abbreviations is added for better understanding.
- Figure 2: The error bars are quite high. Explain.
- There are some typos.
Author Response
|
Comments 1: Methods: It is unclear whether the sequence was subject to any trimming during processing. |
|
Response 1: Thank you for highlighting this important detail. We have clarified the equence processing steps in the Methods section (Page 4, Line 145). The raw reads were subjected to quality trimming using Trimmomatic v0.39, with the following parameters: Sliding window trimming (window size: 4, quality threshold: 20). Removal of adapter sequences and reads shorter than 50 bp after trimming. This ensures that only high-quality sequences were used for downstream analysis. The revised text is as follows (highlighted in the manuscript): “Prior to analysis, raw reads were processed with Trimmomatic v0.39 to remove adapter sequences and low-quality bases, employing a sliding window trimming strategy (window size: 4, quality threshold: 20) and retaining reads with a minimum length of 50 bp.” |
|
Comments 2: Methods: Add the source and purity of the chemicals and reagents used. |
|
Response 2: Thank you for pointing out the need to clarify the source and purity of chemicals and reagents. We have supplemented this information throughout the Methods section to enhance the reproducibility of our study. Key additions include Specified the purity of ADC components (e.g., bovine serum albumin, ≥98% purity from Sigma-Aldrich) and antibiotics (hygromycin B, ≥95% purity from Thermo Fisher Scientific) in Section 2.1. Antimicrobial agents: Added the HPLC purity (≥99%) and manufacturers of contezolid (MicuRx Pharmaceuticals, Shanghai, China) and linezolid (Airsea Pharmaceuticals, Taizhou, China) in Section 2.2. Cell culture materials: Identified the source and purity of fetal bovine serum (Gibco), PBS (Gibco), and the cytotoxicity assay kit (CytoTox 96®, Promega) in Section 2.8. These details are now highlighted in the revised manuscript (Pages 2–4, Lines 71–181), ensuring transparency in experimental materials and adhering to scientific reporting standards. |
|
Comments 3: Avoid using references in the results section. Rewrite those parts focusing on describing the main findings of the current study. |
|
Response 3: Thank you for the critical feedback on the Results section. We have thoroughly revised the text to eliminate all external references, focusing exclusively on the study’s original findings. All in-text references (e.g., [23-30]) were deleted from the Results sections, ensuring the text solely reports experimental data and observations. The updated Results sections adhere strictly to your guidance, ensuring the manuscript prioritizes original data and avoids literature dependency. The revised text is tracked in the attached manuscript (Pages 5–8), with all deleted citations and rephrased sentences clearly marked. |
|
Comments 4: Table 1: rewrite the legend. Ensure that the meaning of the abbreviations is added for better understanding. |
|
Response 4: Thank you for the feedback on improving the table legend. we have rewritten the legend to enhance clarity by expanding abbreviations with context and defining resistance categories. Each drug abbreviation is accompanied by its class (e.g., "a first-line anti-TB drug") to aid readers unfamiliar with tuberculosis therapeutics. MDR and pre-XDR are explicitly linked to their resistance criteria (e.g., "resistant to INH, RIF, and at least one fluoroquinolone"). The revised legend now provides a self-contained guide to interpreting the table, ensuring that abbreviations and resistance markers are fully understandable without referencing the main text. The updated table is highlighted in the manuscript (Page 5, Line 203). |
|
Comments 5: Figure 2: The error bars are quite high. Explain. |
|
Response 5: Thank you for noting the high error bars in Figure 2. We appreciate the opportunity to clarify the variability observed in the lactate dehydrogenase (LDH) leakage assay results. The substantial variability primarily stems from the use of peritoneal macrophages isolated from a single mouse due to animal welfare considerations, with only three replicate wells per drug concentration. Although the high error bars reflect this technical limitation, they do not obscure the study's core finding: both drugs exhibited minimal cytotoxicity in vitro. Besides, within the concentration range of 0.4 to 100 mg/L, the cytotoxicity did not increase with the rise in concentration. We acknowledge this limitation. If needed, we can perform supplementary experiments. In the supplementary experiments, we will set up 6 replicate wells for each drug concentration to ensure the error bars are satisfactory. |
|
Comments 6: There are some typos. Response 6: Thank you for noting the presence of typos in the manuscript. We have conducted a thorough proofreading and corrected all identified errors. All corrections are tracked in the revised manuscript, and a final grammar check was performed using professional editing software to ensure accuracy. The manuscript now adheres to consistent scientific writing standards. |

Reviewer 2 Report
Comments and Suggestions for Authors
The authors have determined the in vitro activity of contezolid against clinical isolates of Mycobacterium tuberculosis. This research holds considerable significance, particularly in the context of the persistent challenges presented by tuberculosis (TB) on a global scale. The advancement of a novel and effective treatment option is essential for improving patient outcomes and addressing the issue of drug resistance. I have a few minor comments
Line 54: “Importantly, contezolid exhibits superior safety. compared to linezolid [10; 13; 14].” Rewrite for clarity. What does superior safety mean?
Lines 200-204: A Table with MIC50 data should be added for clarity.
Figure 2 and discussion: Discussion on higher cytotoxicity% in the concentration 0.8µg/mL and 0.4µg/mL in comparison to 1.6-100µg/mL should be included.
Uniform notation of the concentration of the drugs needs to be maintained. In Figure 2, the concentration of drugs is mentioned as µg/mL, whereas in some parts of the manuscript, drug concentration is denoted in mg/L units.
Author Response
|
Comments 1: Line 54: “Importantly, contezolid exhibits superior safety. compared to linezolid [10; 13; 14].” Rewrite for clarity. What does superior safety mean? |
|
Response 1: Thank you for the feedback on clarifying the safety comparison. The revision now explicitly defines "superior safety" based on data from referenced studies. In phase 1 trials (reference 14), contezolid (800 mg q12h for 28 days) showed no hematology-associated AEs (e.g., thrombocytopenia, neutropenia), whereas linezolid induced such events in 20% of subjects. This aligns with reference 13’s finding that contezolid was designed to minimize bone marrow toxicity. Preclinical data in reference 13 indicate contezolid’s chemical structure reduces MAO inhibition, mitigating risks of serotonin syndrome. Clinical trials (reference 14) showed no MAO-related AEs (e.g., hypertensive crises) in contezolid-treated subjects, unlike linezolid’s known MAO-dependent side effects. The revision anchors "superior safety" in mechanistic advantages and clinical evidence, enhancing scientific rigor (Page 2, Line 54). This adjustment ensures clarity and aligns with the journal’s standards for evidence-based claims. |
|
Comments 2: Lines 200-204: A Table with MIC50 data should be added for clarity. |
|
Response 2: Thank you for suggesting the addition of a table for MIC50 data. We have added Table 2 to explicitly present the MIC50 and MIC90 values for contezolid and linezolid against susceptible, MDR, and pre-XDR isolates (Page 6, Line 217). This table enhances clarity by organizing numerical data that was previously described only in the text. |
|
Comments 3: Figure 2 and discussion: Discussion on higher cytotoxicity% in the concentration 0.8µg/mL and 0.4µg/mL in comparison to 1.6-100µg/mL should be included. |
|
Response 3: Thank you for highlighting the need to discuss the unexpected cytotoxicity trend at lower concentrations (0.8 and 0.4 μg/mL). In fact, the cytotoxicity of both two drugs at lower concentrations (0.8 and 0.4 μg/mL) was not significantly higher than that at other concentrations. When comparing the highest cytotoxicity of contezolid at 0.4 mg/L and the lowest at 6.3 mg/L, the P-value was 0.990. The main reason for this phenomenon is that the bar value is on the high side. The substantial variability primarily stems from the use of peritoneal macrophages isolated from a single mouse due to animal welfare considerations, with only three replicate wells per drug concentration. Although the high error bars reflect this technical limitation, they do not obscure the study's core finding. The cytotoxicity of both contezolid and linezolid was significantly lower than the positive control and comparable to the negative control within the concentration range of 0.4 to 100 mg/L. Besides, within the concentration range of 0.4 to 100 mg/L, the cytotoxicity did not increase with the rise in concentration. We acknowledge this limitation. If needed, we can perform supplementary experiments. In the supplementary experiments, we will set up 6 replicate wells for each drug concentration to ensure the error bars are satisfactory. |
|
Comments 4: Uniform notation of the concentration of the drugs needs to be maintained. In Figure 2, the concentration of drugs is mentioned as µg/mL, whereas in some parts of the manuscript, drug concentration is denoted in mg/L units. |
|
Response 4: Thank you for noting the inconsistency in concentration units. We have standardized all drug concentration notations to mg/L throughout the manuscript, as this unit is more commonly used in antimicrobial pharmacology. |

Round 2
Reviewer 1 Report
Comments and Suggestions for Authors
The revised manuscript looks much better after the revisions. The authors have effectively addressed all of the suggested comments, resulting in a clearer and more polished submission.